# Cost Consequences for the NHS of Using a Two-Step Testing Method for the Detection of *Clostridium difficile* with a Point of Care, Polymerase Chain Reaction Test as the First Step

**DOI:** 10.3390/diagnostics10100819

**Published:** 2020-10-14

**Authors:** William S. Jones, Stephen Rice, H. Michael Power, Gregory Maniatopoulos, Jana Suklan, Fiona Beyer, Mark H. Wilcox, Michelle Permain, A. John Simpson, D. Ashley Price, A. Joy Allen

**Affiliations:** 1NIHR Newcastle In Vitro Diagnostics Co-operative, Room M2.088, Translational and Clinical Research Institute, William Leech Building, Medical School, Newcastle University, Newcastle NE2 4HH, UK; will.jones@ncl.ac.uk (W.S.J.); hmichaelpower@gmail.com (H.M.P.); jana.suklan@ncl.ac.uk (J.S.); j.simpson@ncl.ac.uk (A.J.S.); david.price15@nhs.net (D.A.P.); 2Health Economics Group, Institute of Health and Society, Newcastle University, Baddiley-Clark Building, Richardson Road, Newcastle upon Tyne, NE2 4AA , UK; stephen.rice@ncl.ac.uk; 3NIHR Newcastle In Vitro Diagnostics Co-operative, The Newcastle Upon Tyne Hospitals NHS Foundation Trust, Newcastle upon Tyne NE1 4LP, UK; 4Northumbria University, Newcastle upon Tyne, NE1 8QH, UK; gregory.maniatopoulos@northumbria.ac.uk; 5Population Health Sciences Institute, Faculty of Medical Sciences, Newcastle University, Baddiley-Clark Building, Richardson Road, Newcastle upon Tyne. NE2 4AX, UK; fiona.beyer@newcastle.ac.uk; 6Healthcare Associated Infections Research Group, NIHR Leeds In Vitro Diagnostics Co-operative, Leeds Teaching Hospitals NHS Trust and University of Leeds, Leeds LS1 3EX, UK; mark.wilcox@nhs.net; 7Microbiology and Virology Services, Freeman Hospital, Newcastle upon Tyne, NE7 7DN, UK; michelle.permain@nhs.net; 8Department of Infectious Diseases, Royal Victoria Infirmary, Newcastle upon Tyne, NE1 4LP, UK

**Keywords:** point-of-care testing, *Clostridium difficile*, *C. diff*, health economics, cost-analysis, cost-consequences, infection control

## Abstract

*Clostridium difficile* infection (CDI) is a common healthcare-associated infection. Current practice for diagnosing CDI in the Newcastle upon Tyne Hospitals NHS Foundation Trust involves a three-step, laboratory testing strategy using glutamate dehydrogenase (GDH) enzyme immunoassay (EIA), followed by a polymerase chain reaction (PCR) test then a toxin EIA. However, a PCR point of care test (POCT) for the *C. difficile* tcdB gene for screening suspected CDI cases, may provide a more efficient way of facilitating an equally effective, two-step, testing strategy with a toxin EIA. This study evaluated the cost consequences of changing from the three-step to a two-step testing strategy. A cost-consequences model was developed to compare the costs and consequences of the two strategies. Uncertainties in the model inputs were investigated with one- and two-way sensitivity analysis. The two-step, POCT strategy was estimated to save £283,282 per 1000 hospitalized NHS patients with suspected infectious diarrhea. Sensitivity analysis indicated that the turnaround time for the POCT was the largest driver for cost savings. Providing the POCT has sufficiently high diagnostic accuracy for detecting *C. difficile*, the two-step, POCT strategy for CDI identification is likely to be cost saving for NHS hospitals with an offsite laboratory.

## 1. Introduction

*Clostridium difficile* can be carried without symptoms but is highly transmissible, and in its infectious form (i.e., *C. difficile* infection (CDI)) can result in severe disease and death. UK [1], and European guidelines [2,3], recommend, as a minimum, a two-step testing protocol using laboratory tests: First a screening test with either a glutamate dehydrogenase (GDH) enzyme immunoassay (EIA) or a test for the tcdB gene of *C. diff* encoding for toxin B; then, if the screening test is positive, a toxin assay for toxin B (often an EIA). A third step to test for the tcdB gene may be added for patients who have a positive GDH EIA and negative toxin EIA, to determine whether the patient is a *C. difficile* carrier. Some organizations—including the Newcastle upon Tyne Hospitals NHS Foundation Trust (NuTH)—have an adapted strategy of first testing all samples with a GDH EIA, then, if GDH EIA positive, the sample is tested for the tcdB gene with a polymerase chain reaction (PCR) test; finally, if the PCR is positive, samples are tested for toxin B, via a toxin EIA. This adaptation is justified by a service evaluation performed by the NuTH laboratories, in which they demonstrated that performing PCR prior to toxin EIA testing ensures that any toxin positive results are due to the production of toxin from genes detected. This eliminates the chances of nonspecific reactions detected in the toxin EIA test being falsely reported as an active CDI and the cause of diarrhea (see Appendix A for more details of this evaluation). Hereafter, the adapted NuTH strategy will be referred to as the *laboratory testing strategy*.

Point of care tests (POCTs) offer the ability to perform a diagnostic test outside of the laboratory setting, near to the patient with rapid results to inform patient management in real-time. POCTs [4], using PCR technology to detect the tcdB gene, may allow CDI to be diagnosed more efficiently (e.g., by running the assay within the ward or in a medical assessment unit) and no less accurately than current practice [5]. Furthermore, a PCR POCT potentially allows for the rapid exclusion of a CDI diagnosis. Therefore, these tests could facilitate more appropriate infection control and patient management, optimizing the use of NHS resources. This work aims to assess the potential cost and consequences to the NHS of adopting a rapid POCT for the tcdB gene, as part of a two-step diagnostic approach.

## 2. Materials and Methods

### 2.1. Diagnostic Strategies

A decision model was developed to compare two strategies for diagnosing CDI in hospital patients with suspected infectious diarrhea (see Appendix A). The laboratory testing strategy is the protocol described in the introduction, which uses a three-step approach with laboratory tests. The POCT strategy is a two-step approach in which samples are first tested with a PCR POCT for the tcdB gene, in this study, the Revogene™ *C. difficile* assay (Meridian Bioscience, Cincinnati, OH, USA). Positive samples are then tested with a toxin EIA, in the laboratory. See Figure 1, below, for an outline visual representation of diagnostic testing strategies. More detailed clinical care pathways for the two strategies can be found in Appendix A.

The model assumes that, with both strategies, the patients are adults admitted to or are already in hospital (hereafter referred to as ‘hospitalized patients’) with suspected infectious diarrhea. If the clinical team suspects a CDI, the patient will be presumptively placed in side-room isolation whilst a stool sample is collected and laboratory results returned—that is, assuming a side room is available, which is not always the case. With both strategies, the stool is routinely tested for a panel of pathogens known to cause infectious diarrhea. However, this routine is subject to alteration if the patient has been in hospital for >3 days, in which case, it is considered unlikely that the patient has a community-acquired infective diarrhea.

In the laboratory testing strategy, the sample undergoes the three-step testing for CDI. In the newly-proposed POCT strategy, the sample is tested in a ward or a medical assessment unit using the Revogene™ *C. difficile* assay, and if positive, is further tested for presence of toxin B, with the toxin EIA, based in the laboratory (see Figure 1. Simplified visual representation of diagnostic testing strategies examined in this study for visual representation of testing strategies). With both testing strategies, if the toxin result is positive, the patient will be managed for CDI, and if the toxin result is negative, the patient will be discharged if well or will continue in-patient management. These clinical care pathways were mapped and validated through 13 semi-structured interviews with expert clinicians and laboratory staff (see Appendix A for more details).

### 2.2. Analysis

For utility of reporting and ease of approximate generalization to NHS hospitals we used a hypothetical cohort of 1000 patients. The time horizon for analysis was two weeks, which covers diagnosis, management, and infection control procedures. Costs were estimated from the perspective of the UK NHS and are presented in pounds sterling for 2017/2018. The outcomes of the decision model included time to result, time in side-room isolation, total length of hospital stay, cost of bed days, costs of diagnostic testing, and total costs; details are in Appendix A.

### 2.3. Model Parameters

We searched for existing economic models addressing the diagnosis or care of patients with *C. difficile* or diarrhea. We searched MEDLINE and EMBASE from inception to March 2018, using thesaurus headings for *C. difficile* and diarrhea combined with a published filter [6]. The search strategy is fully described in Appendix A. Studies were downloaded to Endnote, de-duplicated, and screened. Where studies appeared to contain data relevant to the model inputs, the full text was obtained and scrutinized.

The two strategies involve tests being performed sequentially. The diagnostic accuracies of these tests, as compared with toxigenic culture, were used to calculate the predictive probabilities, which determine the proportion of patients for each diagnostic classification.

The time to result for all tests in the laboratory testing strategy were estimated with a workflow analysis in the NuTH microbiology laboratory. The accuracy statistics for these tests were extracted from a recent meta-analysis [2]. The accuracy statistics and time to result for the Revogene™ *C. difficile* assay of the POCT strategy were extracted from the product information on the Revogene™ *C. difficile* assay website [7]. These inputs are summarized in Appendix A.

List costs for the laboratory tests were obtained from the NuTH microbiology laboratory. These costs do not take into account the cost of laboratory staff to run the tests. The cost of the POCT for the tcdB gene is the Revogene™ *C. difficile* assay is its list price and does not take into account the cost of clinician time to run the test.

Costs for bed-days were obtained from NHS Reference Costs 2015/2016 [8] and inflated to 2017 prices using the Bank of England Inflation Calculator [9]. For stays in an open ward, this cost was an average based on patients with “gastrointestinal infections without interventions,” which was obtained from the elective inpatient spreadsheet [8]. For patients who stayed in isolated beds, an additional cost of £96.50 per day was added [10]. For each day a patient spends in hospital a cleaning cost is added to account for staff time, specialist cleaning equipment, and the cost of disposables. Unit costs for these resources were taken from NHS Reference costs 2015/2016, inflated to 2017 costs are detailed in Appendix A.

The common causes of infective gastroenteritis, the proportion of cases which are treated, and for each possible treatment, the probability of that treatment being prescribed are detailed in Appendix A. Treatment is typically not recommended for shiga toxin-producing E. coli O157. There is no recommended treatment in the UK NHS for cryptosporidium, therefore no costs have been included. Treatment costs are summarized in Appendix A.

### 2.4. Exploring Uncertainties in Model Outputs

To assess uncertainties in the results and to identify the input parameters that have the greatest effects on the model’s results, we conducted one- and two-way deterministic sensitivity analyses using TreeAge^®^ Pro 2016 [11]. For the two-way sensitivity analysis we chose to vary the prevalence of both CDI and other gastro pathogens from 0 to 25% to reflect increasing prevalence within epidemic situations and potential changes with seasonality [12]. Time to obtaining sample was varied between 0 and 2 days as this parameter is known to vary widely for multiple reasons. Likewise, the transport time of a sample to the laboratory can vary widely among UK hospitals so this parameter has been varied between 0 and 1 days. The cost of bed days (in side-room isolation and in an open ward) has been varied by ±50%. All other parameters were varied between plausible ranges elicited from the clinical experts consulted in the interviews. The ranges used are fully outlined and can be found in Appendix A.

A scenario analysis explored the effect on model outcomes if patients who test positive with a tcdB gene test and negative with a toxin EIA (i.e., are *C. difficile* carriers) were to be treated in the same way as patients who test positive with a toxin EIA.

## 3. Results

### 3.1. Systematic Search for Previous Health Economic Models

One study was found to be relevant to the model structure of this research; the Freeman et al. health technology assessment [12] and was used to aid model development. For full details of search see Appendix A.

### 3.2. Accuracy of Test Strategies

For a cohort of 1000 patients with diarrhea, the laboratory testing strategy classifies 54 patients as true positive (TP) for CDI, zero patients as false positive (FP), 897 as true negative (TN), and 50 as false negative (FN). The POCT strategy classifies 56 as TP, 1 as FP, 896 as TN, and 48 as FN.

### 3.3. Time Outcomes

Overall, the POCT strategy is estimated to produce quicker time to actionable results (0.89 days vs. 1.35 days) and reduces the total length of stay per patient by 0.21 days (Table 1). For both testing strategies, patients who are presumptively isolated spend longer in side-room isolation than those who are not initially placed in isolation, and longer for the laboratory testing pathway than the POCT strategy.

### 3.4. Cost Outcomes

The POCT strategy is estimated to provide total savings of £283,282 for a cohort of 1000 hospitalized patients with suspected infectious diarrhea. The laboratory testing strategy is less costly than the POCT strategy for both diagnostic testing costs and treatment costs, however, is significantly more costly in terms of cost of total bed days (Table 2).

The sensitivity analysis showed that the prevalence of CDI in patients with diarrhea was found to be the most influential parameter on per-patient expected cost of the POCT strategy, with increased prevalence resulting in increased costs. The time to obtain a sample followed by cost of side-room isolation (per-day) are the next most influential parameters. The cost of a day on an open ward and length of stay in hospital after CDI detected are next, followed by the time to obtain a sample and the time in hospital for CDI and the time to result of the POCT. The tornado diagram, presented in Appendix A, illustrates the effect of varying these parameters on the per-patient expected cost of the POCT strategy. Varying the sensitivity and specificity of the POCT had little effect on the cost outcomes of the model. However, increased costs are incurred in the POCT strategy as the time to result of the POCT is increased.

A two-way sensitivity analysis between the transport time to the laboratory and the result turnaround time for the POCT is shown in Figure 2. Two-way sensitivity analysis on the transport time to the laboratory (days) and the time to POCT result (days). Results presented are per-patient expected values for both testing strategies.

The dividing line indicates equal costs between the two strategies. This figure demonstrates that with increased time to result for the POCT and decreased sample transport time to the laboratory, the laboratory testing strategy is the less costly of the two. This has implications for hospitals with a laboratory in the immediate vicinity of the ward or medical assessment unit.

Further analysis to explore the cost implications of treating *C. difficile* carriers in the same way as CDI positive patients, shows that POCT strategy remains the less costly testing strategy (£1,442,360 vs. £1,883,550), however, it is at a lower cost in this scenario than in the base case. The largest contribution to the cost remains the cost incurred from total bed days, both those in side-room isolation and on an open ward.

## 4. Discussion

The results from this cost-consequence model indicate that the two-step, *POCT strategy* was estimated to save £283,282 per 1000 hospitalized NHS patients with suspected infectious diarrhea. Sensitivity analysis indicated that the turnaround time of the POCT was the largest driver for cost savings. The largest contribution to costs of both strategies was the length of stay in hospital (both in side-room isolation as well as on an open ward), indicating that a decreased time to actionable results is likely to reduce the overall cost of hospital stay for these patients. The cost of each strategy was highly dependent on the prevalence of CDI, with increased prevalence resulting in increased costs.

That turnaround time was the key driver of cost savings suggests that hospitals which have laboratories situated on-site (i.e., close to the medical assessment units with regular transportation of samples) may not save costs by adopting a POCT. However, hospitals with longer transport time may save costs by moving to a *POCT strategy*, provided the turnaround time of the result is low, which would need to be demonstrated in a real-life, clinical setting.

Finally, the above analysis revealed that if a POCT has sufficiently high diagnostic accuracy, then the *POCT strategy* may result in an increased negative predictive value, in comparison to the *laboratory testing strategy*. This is a valuable characteristic of a diagnostic employed in a disease area with low prevalence, such as *C. difficile*.

### Limitations

The analysis presented has assumed that a hospitalized patient with suspected infectious diarrhea will either be isolated or not. However, in reality if side-room isolation is unavailable, the clinical team will invoke other infection control procedures, such as barrier nursing and personal protective equipment—these costs have not been included in the model. Other costs, such as diagnostic procedures, spot cleaning costs, routine blood tests carried out on admission, routine blood chemistry and C-reactive protein tests, sigmoidoscopy test costs have also not been included.

Due to the short time horizon, we have not included the cost of onward transmission of CDI from patients who are CDI positive or the cost of longer-term consequences associated with CDI.

In the *laboratory testing strategy*, utilized by the NuTH labs, their PCR test also detects the binary toxin gene, which together with the tcdB gene can be an indicator of more severe disease or recurrence. This additional test is not considered in this model. However, the clinical significance of this test information is still uncertain.

There are several possible forms and causes of infectious diarrhea, beyond *C. difficile*. Therefore, a negative *C. difficile* result does not mean that the patient’s diarrhea is not infective (e.g., the patient could have a norovirus infection). Consequently, a single negative PCR POCT result may not be sufficient information to remove a patient from side-room isolation. The semi-structured interviews (see Appendix A) revealed that infectious disease consultants were more likely to deisolate on the basis of a *C. difficile* result (only), in comparison to consultant microbiologists, who would not deisolate. This illustrates a challenge faced by all POCTs for single pathogens, where multiple infective diseases are on the differential and cannot be ruled out.

Finally, as stated in the Introduction, the *laboratory testing strategy* in this study is an adapted version of recommended guidelines. Therefore, subsequent generalizations made on the basis of the results in this paper should be mindful of these differences.

## 5. Conclusions

The aim of this study was to model the economic consequences of adopting a PCR POCT for the detection of *C. difficile*, as part of a two-step, *POCT strategy*, paired with a laboratory toxin EIA test. This was in comparison to the three-step, *laboratory testing strategy*, of GDH EIA, followed by PCR, followed by a toxin EIA—current practice in the NHS NuTH laboratories. The results indicated that a shift to a *POCT strategy* for CDI identification in UK NHS hospitals is likely to be cost saving for a hospital without a laboratory in the immediate vicinity of the ward or medical assessment unit.

Future research should explore the costs and accuracies of the different *C. difficile* testing strategies employed across UK NHS laboratories. Also, to establish the true value of a PCR POCT for *C. difficile*, it is important to understand the complex decision-making process involved in *C. difficile* diagnosis. In particular, in what cases would a negative PCR POCT for *C. difficile* facilitate de-isolation? Based on the qualitative research performed in this study (see Appendix A), it was concluded that deisolation may only be appropriate in the subgroup of patients where other non-infective causes of diarrhea can be ruled-out. Further work is required to confirm this proposition. Finally, future research should evaluate the clinical outcomes of adopting a POCT for identifying *C. difficile*. Does the quicker result from the POCT result in earlier and more effective treatment in patients with confirmed *C. difficile*? Does it result in a more appropriate use of isolation resources? Are there potential harms?

## Figures and Tables

**Figure 1 diagnostics-10-00819-f001:**
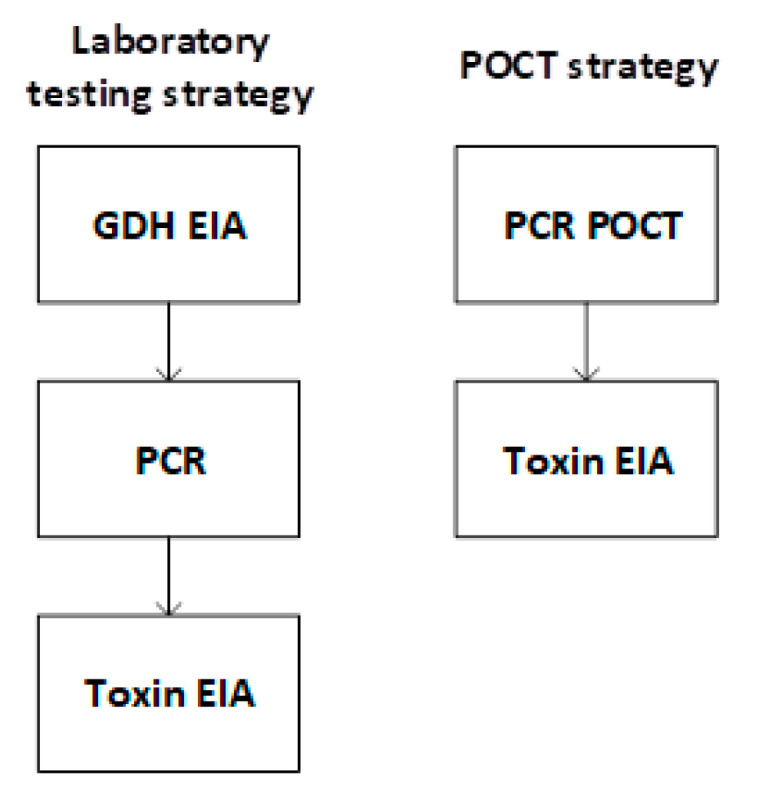
Outline visual representation of the diagnostic testing strategies examined in this study.

**Figure 2 diagnostics-10-00819-f002:**
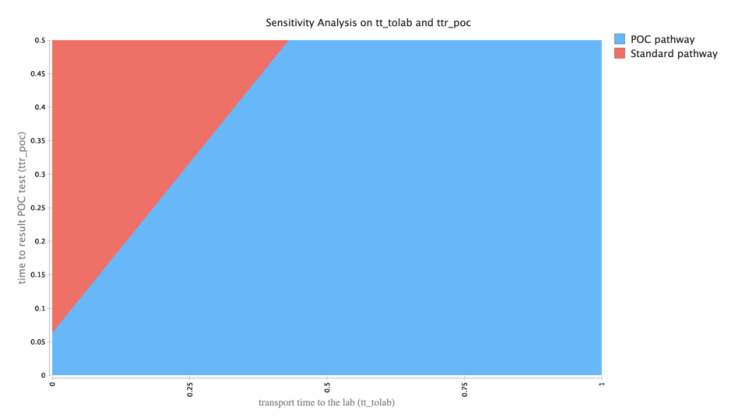
Two-way sensitivity analysis on the transport time to the laboratory (days) and the time to POCT result (days). Results presented are per-patient expected values for both testing strategies.

**Table 1 diagnostics-10-00819-t001:** Time analysis of laboratory testing and point of care test (POCT) strategies for a cohort of 1000 hospitalized patients with suspected infectious diarrhea. The results presented are an average, per patient value for isolated and non-isolated patients. Standard deviations are presented in brackets.

	Laboratory Testing Strategy (Days)	POCT Strategy (Days)	Incremental Time (Laboratory Testing Strategy—POCT Strategy) (Days)
**Patients Starting out in an Isolation Bed (*n* = 511)**			
**Time in isolation bed**	1.33	0.80	0.53
**Total length of stay**	1.37	1.16	0.21
**Patients initially admitted to a general bed (*n* = 489)**			
**Time in isolation bed**	0.62	0.64	−0.02
**Total length of stay**	1.32	1.11	0.21

**Table 2 diagnostics-10-00819-t002:** Cost analysis of the laboratory testing and POCT strategies for a cohort of 1000 hospitalized patients with suspected infectious diarrhea.

Cost of…	Laboratory Testing Strategy	POCT Strategy	Incremental Cost(Laboratory Testing Strategy—POCT Strategy)
**Patients starting out in an isolation bed (*n* = 511)**			
**Total bed days (isolation + general)**	£909,845.72	£734,674.92	£175,170.80
**Diagnostic testing costs**	£6372.17	£11,589.48	−£5217.31
**Treatment**	£5329.73	£5610.78	−£281.05
**Total costs**	£919,728.46	£751,864.96	£167,863.50
**Patients starting out in a general bed (*n* = 489)**			
**Total bed days (isolation + general)**	£804,004.02	£690,375.09	£113,628.93
**Diagnostic testing costs**	£15,393.72	£11,085.63	£4308.09
**Treatment**	£5100.27	£5369.22	−£268.95
**Total costs**	£825,236.40	£709,817.73	£115418.67
**Total cost for cohort**	£1,744,964.86	£1,461,682.90	£283,282.17

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
