# Peer review of "Cost Consequences for the NHS of Using a Two-Step Testing Method for the Detection of Clostridium difficile with a Point of Care, Polymerase Chain Reaction Test as the First Step"

_diagnostics, 2020, doi:10.3390/diagnostics10100819_

Round 1

Reviewer 1 Report

The authors analyzed the economic impact of adopting a point-of-care PCT test for Clostridium difficile infection (CDI). They ran analyses on the overall healthcare costs, individual factors contributing to the care and the cost, and diagnostic accuracy. Compared to the three-step approach in the conventional laboratory testing strategy, the two-step POCT strategy suggested could reduce the overall costs via reduced hospitalization duration. There is no significant difference in terms of diagnostic accuracy. The cost reduction benefit would be greater for a hospital without a laboratory that needs to transport samples for lab testing. 

While there is no surprising new result in the study, it basically demonstrates the economic power of a POCT strategy in a clinic with limited resources. The analysis model was carefully designed and analyzed. The manuscript is well written and the conclusion is supported well with the results. I have a couple of minor comments:

  1. In Table 1, there is no information about the standard deviation of the time durations and patient numbers in isolation and general beds, respectively. I also suggest the authors run statistical tests for significance between laboratory and POCT strategies.
  2. Line 161, 172: please fix the reference errors
  3. Table 2: correct "in an general bed"

Author Response

We thank the reviewer for their helpful comments. We address these below. 

1. While we agree this would be helpful information, it is inappropriate for this analysis. This is an economic evaluation that uses an economic model to estimate the expected incremental time in hospital and the expected incremental cost. It's not an individual patient data analysis and so no statistical test can be conducted. Furthermore, we undertook a deterministic analysis and in such an analysis uncertainty can only be evaluated through one-way, two-way etc. sensitivity analysis. Had a probabilistic sensitivity analysis been conducted we could have calculated the probability that POC resulted in shorter hospital stay or was cheaper, but that's not the case.

We have added patient numbers to Table 1 and 2. 

2.  Apologies, the references should be fixed in the uploaded version. 

3.  Thank you for pointing this typo out, we have fixed in the uploaded version. 

Reviewer 2 Report

This manuscript showed interesting results on the difference between POCT and lab testing for CDI. The following comments to help the authors in improving the manuscript before an acceptance.

  1. Line 160: (Error! Reference source 162 not found.)
  2. Line 172: (Error! Reference source not found.)
  3. The authors have not defined what POCT means in this manuscript. A clear definition of point of care testing can be essential for the readers. An example of POC testing definition can be found in https://doi.org/10.1016/j.trac.2020.116004

Author Response

We thank the reviewer for reading the manuscript so carefully.  

  1. The references should now be fixed in the uploaded version. 
  2. As above
  3. 3.  We thank the reviewer for this observation and drawing our attention to the reference.  We  have added a definition in the introduction and this reference. 

Round 2

Reviewer 2 Report

There might be a mistake in the citation/reference in the manuscript. 'Point of care tests (POCTs) offer the ability to perform a diagnostic test outside of the laboratory 56 setting, near to the patient with rapid results to inform patient management in real-time.' The authors may need to put a reference here at the end of this statement.

Author Response

To the reviewer,  apologies, I didn't upload the latest version of the manuscript. The reference should be in place now.

Thanks for noticing. 

Round 3

Reviewer 2 Report

The authors have addressed my comments.